# Enzymatic Formation of Recombinant Antibody-Conjugated Gold Nanoparticles in the Presence of Citrate Groups and Bacteria

Maryam Rad [1], Gholamhossein Ebrahimipour [1], Mojgan Bandehpour [2,3,*], Omid Akhavan [4,*] and Fatemeh Yarian [2]

1   Department of Microbiology and Microbial Biotechnology, Faculty of Life Sciences and Biotechnology, Shahid Beheshti University, Tehran 1983969411, Iran
2   Cellular and Molecular Biology Research Center, Shahid Beheshti University of Medical Sciences, Tehran 1481997791, Iran
3   Department of Medical Biotechnology, School of Advanced Technologies of Medicine, Shahid Beheshti University of Medical Sciences, Tehran 9168917313, Iran
4   Department of Physics, Sharif University of Technology, Tehran 11155-9161, Iran
*   Correspondence: m.Bandehpour@sbmu.ac.ir or Bandehpour@gmail.com (M.B.); oakhavan@sharif.edu (O.A.); Tel.: +98-021-88666139 (M.B.); +98-21-66164566 (O.A.); Fax: +98-21-66022711 (O.A.)

**Abstract:** With the spread of deadly diseases worldwide, the design of rapid tests to identify causative microorganisms is necessary. Due to the unique properties of gold nanoparticles, these nanoparticles are used in designing rapid diagnostic tests, such as strip tests. The current study aimed to investigate the ability of gold nanoparticles to bind to single-chain variable fragment antibodies. In this study, the biological and chemical methods included *Escherichia coli* TOP-10 and the Turkevich method to synthesize the gold nanoparticles, respectively. Then, the effect of synthetic nanoparticles on their capability of binding to recombinant antibodies was assessed by agarose gel and UV-vis spectroscopy. Our result showed that gold nanoparticles had a spherical morphology, and their average size was ~45 nm. Additionally, the citrate groups in gold nanoparticles were able to bind to serine residues in the antibody linker sequence; so, the chemical synthesis of gold nanoparticles is an effective strategy for binding these nanoparticles to antibodies that can be used in designing rapid diagnostic tests to promptly identify infectious microorganisms.

**Keywords:** nanomaterials; enzymes; green synthesis; antibody conjugation; Au nanoparticles; plasmon resonance

## 1. Introduction

The lateral flow assay is based on antigen and antibody reactions, followed by the generation of red color in a few minutes, which can be detected with the naked eye [1]. Nanoparticles have unique properties that can increase the sensitivity of sensors in two ways—namely, an increase in the affinity to the desired target or an increased signal strength in the optical and electronic properties of nanoparticles. Gold nanoparticles (AuNPs) are used to amplify the signal [2].

The resonant property of nanoparticles causes the oscillation and dispersion of the conducting electrons in gold nanoparticles, which vary depending on their morphology, size and refractive index [3]. By means of the attachment of a layer of biological materials, such as DNA, protein or lectin, to gold nanoparticles, they potentially become bioactive particles [4,5]. In order to use nanoparticles to identify the molecules, it is necessary to coat nanoparticles with specific antibodies. The electron density and unique optical properties of gold nanoparticles, along with the immunological properties of the antibodies, lead to practical applications of AuNPs for disease detection. Compared to the use of a single label, the application of antibodies labeled with gold nanoparticles, together with the

addition of a marker or fluorescent enzyme to the reaction medium, increases the diagnostic signal 10 times. In fact, the gold nanoparticle-antibody complex is particularly effective in detecting low concentrations of small molecules [4].

The method of binding gold nanoparticles to antibodies comprises electrostatic, hydrophobic or direct covalent bonding [3]. Susan van der Heide and David A. Russel (2016) described three strategies for covalent binding of gold nanoparticles to antibodies using polyethylene glycol (PEG)PEG–COOH and PEG–NH2 linkers or the connection through protein A/G with the aid of a N-succinimidyl 3-(2-pyridyldithio)-propionate (SPDP) linker. PEG–COOH can bind to the surface of gold nanoparticles by the citrate group, while PEG–NH2 is able to bind to polysaccharide residues of the FC antibody region to bind gold nanoparticles to the antibody. In the third method, the binding process is mediated by the protein A/G, which binds to the FC region of the antibody, exposing the Fab portion to bind to the target antigen. According to their results, the third method is the most effective way to bind gold nanoparticles to antibodies [4].

The disadvantages of physical adsorption in functionalized nanoparticles include the random orientation of biomolecules on the surface of nanoparticles, which leads to inefficient identification and weaker bonds between the nanoparticles and biomolecules. In biotechnology methods, de novo protein linkers are designed to identify specific areas on the surface of nanoparticles. Liu et al. used genetically engineered linkers to bind recombinant antibodies to gold nanoparticles. Their results showed that nanoparticles attached to single chain variable fragment-cysteine (scFv-cys) have a high sensitivity and acceptable properties [6].

In this study, an enzymatic synthesis of scFv recombinant antibody-conjugated Au NPs was studied in the presence of bacteria and also under a common chemical condition for better comparison of the binding potential of the nanoparticles to the antibodies.

## 2. Results

### 2.1. Characterizations of Nanoparticles

#### 2.1.1. X-ray Diffraction (XRD)

The crystalline structure of the gold nanoparticles was analyzed by X-ray. The indices (111), (200), (220) and (311) correspond to the angles of 38.1°, 44.3°, 64.5° and 77.7° of the synthesized gold nanoparticles, respectively. This result confirms the biosynthesis of the multi-crystal cubic structure of gold nanoparticles [7] (Figure 1).

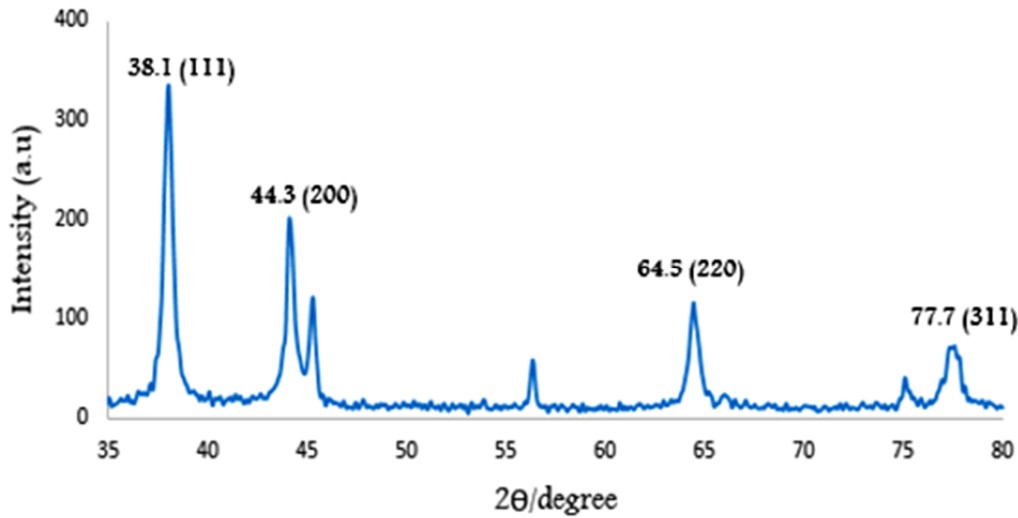

**Figure 1.** The XRD pattern of the synthesized gold nanoparticles.

#### 2.1.2. Field-Emission Scanning Electron Microscope (FESEM)

FESEM detects the morphology and size of nanoparticles and also determines the particle size below 100 nm [8]. As depicted in Figure 2, images obtained from the FESEM

method show that gold nanoparticles have a spherical morphology, and their particle size is in the range of 45–85 nm.

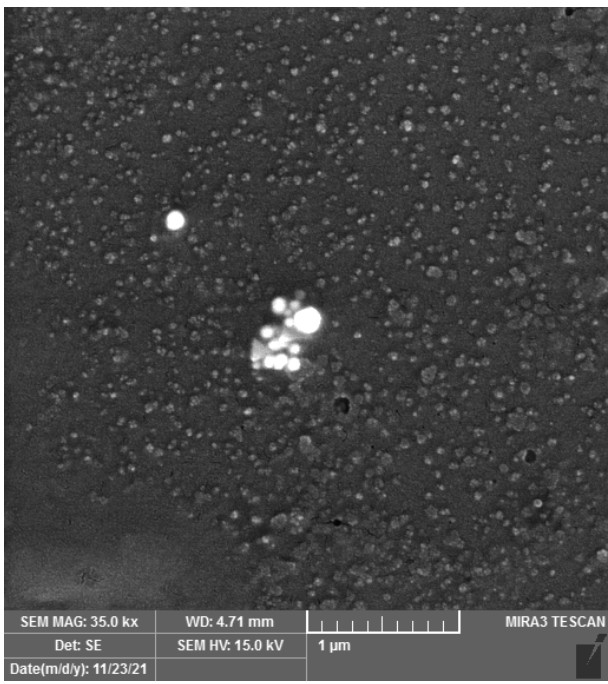

**Figure 2.** FESEM image of Au NPs synthesized by *E. coli* TOP-10.

### 2.1.3. AFM Analysis

Figure 3 shows the size distribution of the synthesized gold nanoparticles. Gold nanoparticles with sizes of 30.4 nm were used for chemical synthesis.

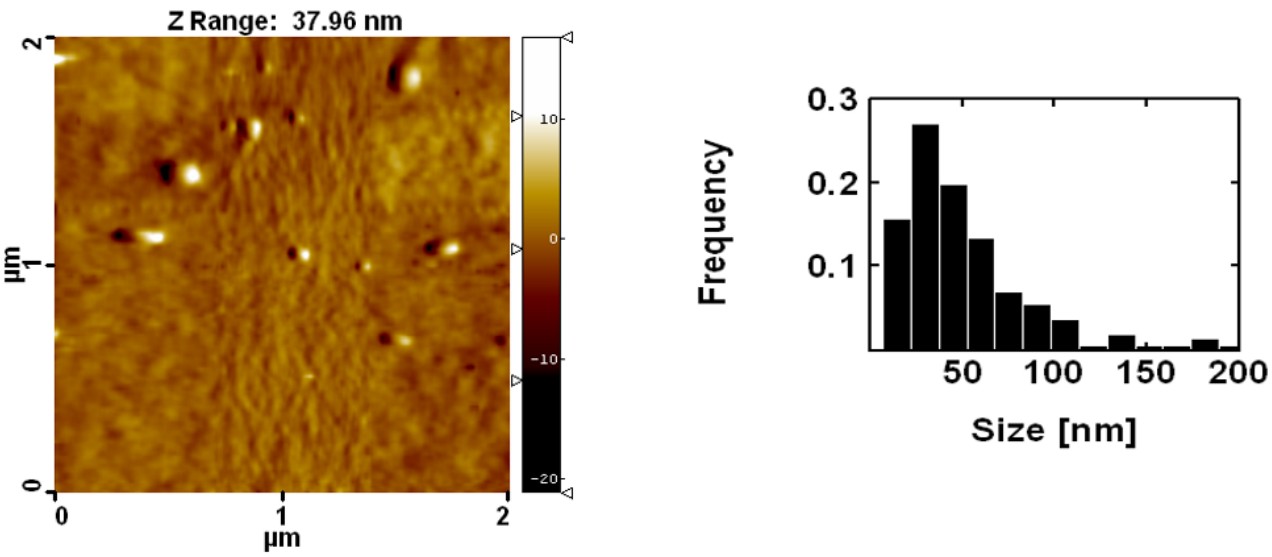

**Figure 3.** AFM image and size distribution of AuNPs prepared by chemical synthesis.

### 2.2. Binding of Gold Nanoparticles to Antibodies

As displayed in Figure 4, after binding to the antibodies, the chemically synthesized nanoparticles show slow motion on the agarose gel, implying the size increment and binding affinity of the recombinant antibodies to gold nanoparticles. However, in the biosynthesized nanoparticles, no difference is observed in the agarose gel before and after the addition of the antibody. Figure 5 shows the change in the absorption peak of chemically

synthesized gold nanoparticles after being conjugated with antibodies (additionally, the gold nanoparticles' plasmon resonance is illustrated in Refs [9,10]). The biosynthesized nanoparticles have the same UV-vis pattern before and after the addition of the antibody.

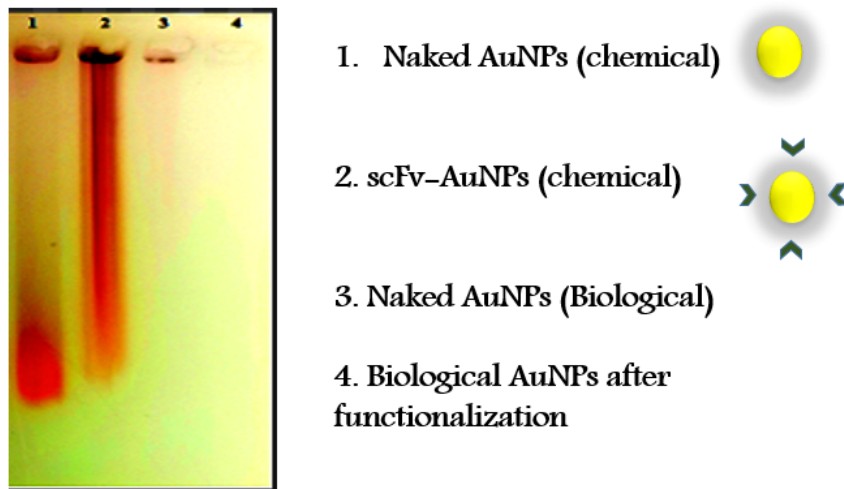

**Figure 4.** Illustrated immigration pattern of AuNPs in various conditions: **1** the chemically synthesized AuNPs, **2** the scFv-conjugated AuNPs prepared by the chemical method, **3** the biologically synthesized AuNPs and **4** the scFv-conjugated AuNPs prepared by the biological method. For the chemical method, the immigration pattern of AuNPs changes, while for the biological method, the patterns are the same before and after functionalization.

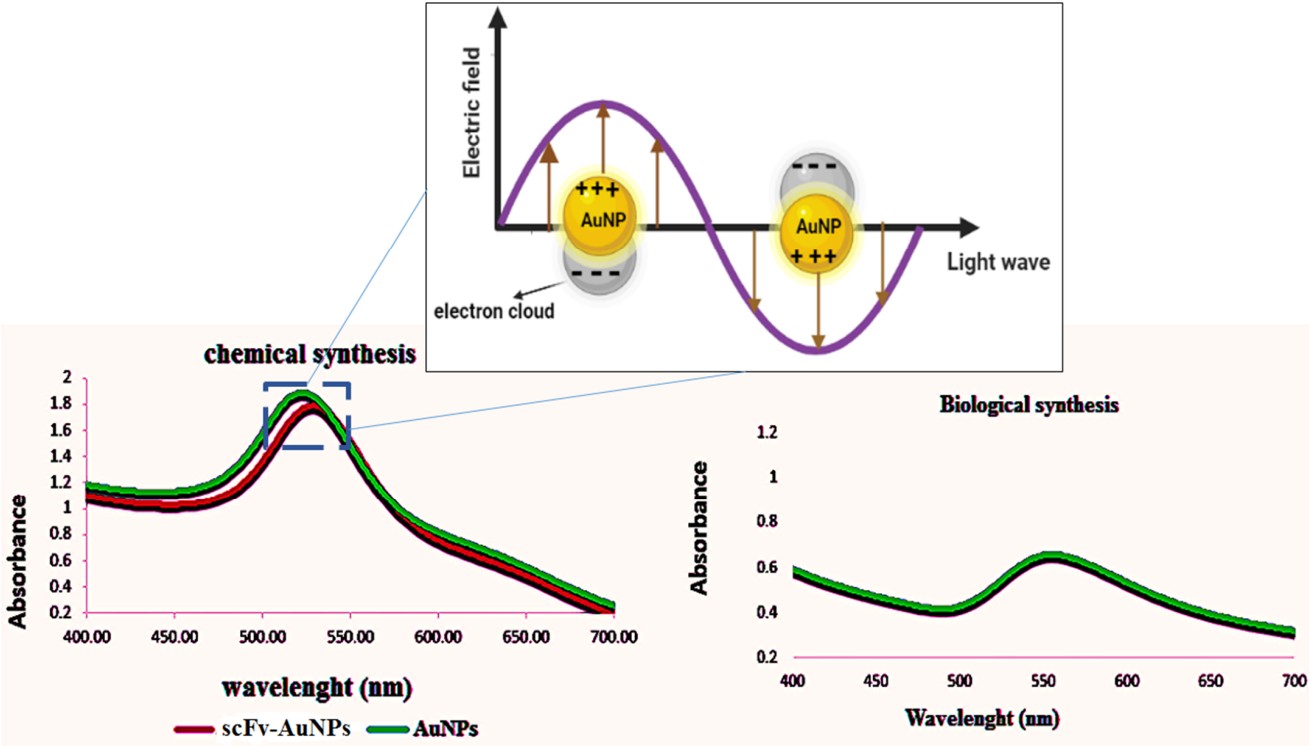

**Figure 5.** The absorption peak of chemically synthesized AuNPs changed after binding to scFv. The same peak before and after reaction for bacterial biosynthesized nanoparticles.

## 3. Discussion

　　　Gel electrophoresis is a method for separating charged nanoparticles that differ in size, shape and charge [11,12]. Therefore, in this study, the aim of investigating the binding

of gold nanoparticles to antibodies on agarose gel was to analyze the binding efficiency of nanoparticles to antibodies, which was different based on the size of nanoparticles before and after the binding process. Additionally, due to the alteration in the maximum absorption peak of gold nanoparticles after conjugation with the antibody [6,13–15], the UV-vis analysis was used to confirm the binding of nanoparticles to the antibody.

The influence of the type of amino acids used in the linker on the binding affinity of gold nanoparticles to antibodies [6], the role of stabilizing agents in preventing the agglomeration of nanoparticles, the stabilizing effect of stabilizers on the interaction of nanoparticles with the synthetic environment, the special structural properties of nanoparticles and their impact on the biological activity of these particles, physicochemical and biological changes of nanoparticles as a result of steric effects of these covering factors [16] and the difference in the affinity of antibodies to gold nanoparticles are all factors that could affect the tendency of serine groups (Figure 6) existing in the linker to bind to the surface citrate of chemically synthesized gold nanoparticles.

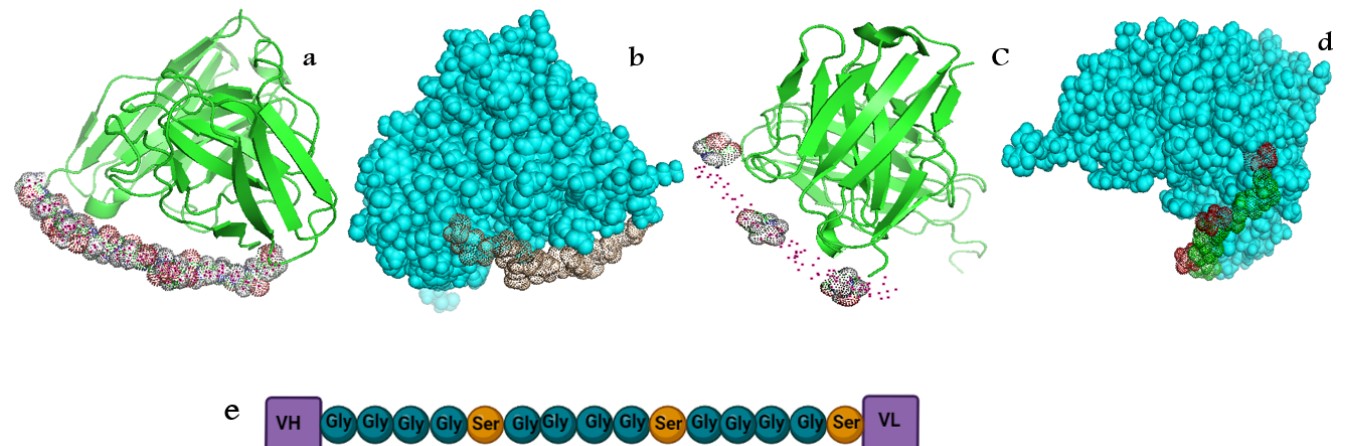

**Figure 6.** (**a**,**c**): two-dimensional structure of scfv. (**b**,**d**): spatial structure of scFv. In these structures, linkers are illustrated with different colors, and bold regions in (**c**,**d**) are serine amino acids. (**e**): schematic structure of scFv.

According to Scheme 1 [17], NH₂ groups of serine residues in the linker sequence are capable of binding to oxygen trisodium citrate. Therefore, based on Scheme 1, the bond between the surface citrate of gold nanoparticles is chemically synthesized, and the serine residues in the antibody are shown in Figure 7.

$$NH_2 + O \longrightarrow HNO + H \qquad \Delta H = -124 \text{ kJ/mol}$$
$$\longrightarrow NH + OH \qquad \Delta H = -43 \text{ kJ/mol}$$
$$\longrightarrow H_2O + N \qquad \Delta H = -211 \text{ kJ/mol}$$
$$\longrightarrow NO + H2 \qquad \Delta H = -351 \text{ kJ/mol}$$

**Scheme 1.** This reaction is possible according to the reaction energy of the bond between nitrogen and oxygen.

In the synthesis of nanoparticles by bacteria, factors such as protein and special compounds that are specific to those bacteria function as a AuNPs capping and stabilizing agent [18] (Figure 8) [18,19]. In general, bacteria use various bio-reduction processes in both intracellular and extracellular pathways for the synthesis of nanoparticles. Intracellular enzymes and positively charged groups act by taking metal ions from the environment and subsequently reducing them inside the cell in the intracellular pathway, but in the

extracellular pathway, NADH-dependent enzymes play a significant role [20]. A study by Susan van der Heide et.al (2016), before using protein A/G as a functionalizing agent, has linked it to SPDP. Accordingly, perhaps the reason for the lack of compatibility observed in the binding of biosynthesized nanoparticles to bacteria in this research was due to the native protein in bacteria not binding with the linker.

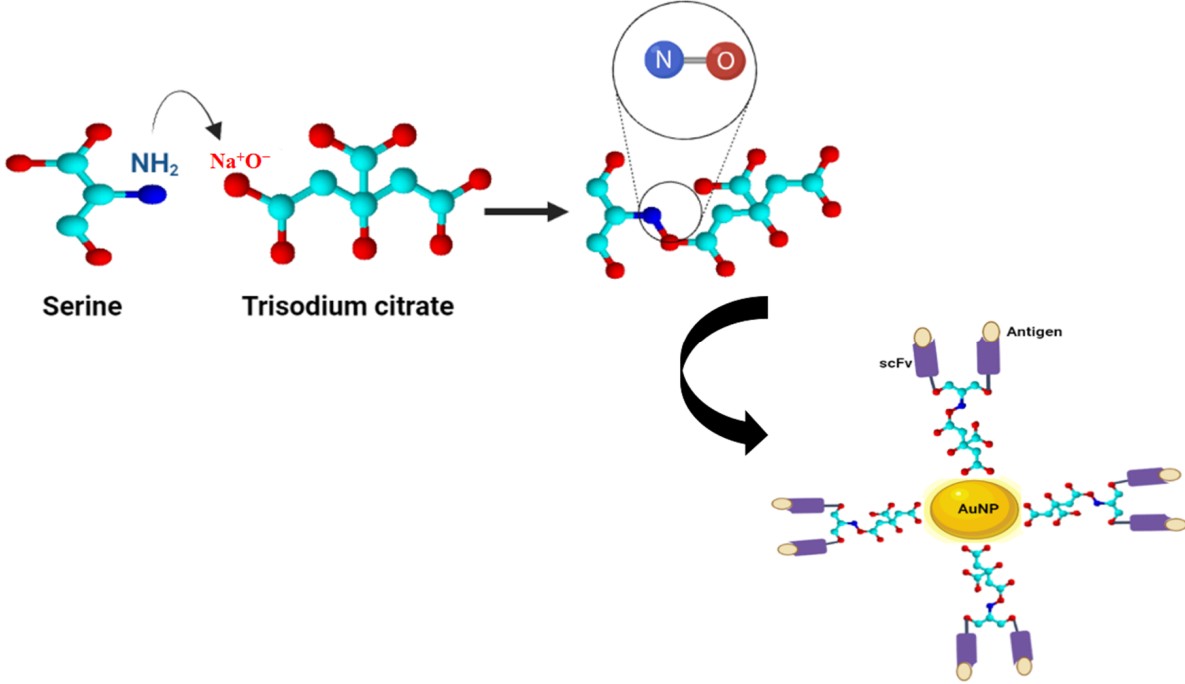

**Figure 7.** Schematic of the binding of serine amino acid to citrate and final situation of product on the gold nanoparticle surface.

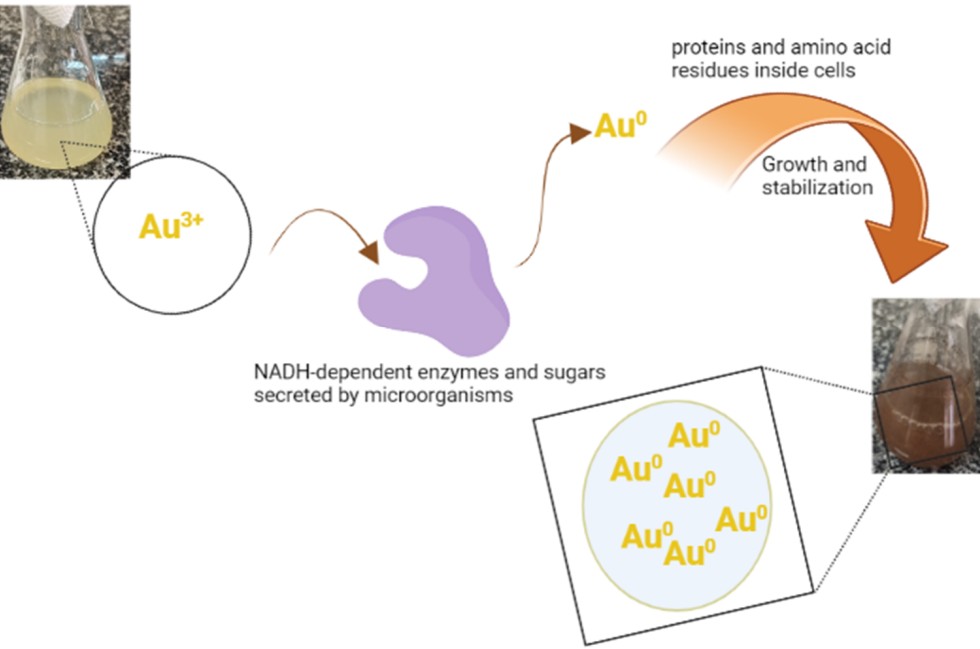

**Figure 8.** Schematic of synthesis, growth and stabilizing agents of biosynthesized nanoparticles.

## 4. Materials and Methods

Chloroauric acid ($HAuCl_4$), trisodium citrate, agarose and nutrient broth were purchased from Sigma-Aldrich (3050 Spruce St, St. Louis, MO, USA).

### 4.1. Biological and Chemical Synthesis of Gold Nanoparticles (AuNPs)

In this experiment, a 50 mL pellet was obtained by centrifugation (1000 rpm, 15 min) from the overnight cell culture of *Escherichia coli TOP-10* (Catalog number: C404010) in the nutrient broth (NB) culture medium. Then, 10 mL of LB broth culture medium was added and pipetted to homogenize. The suspension was sonicated and then centrifuged. After that, 70 μL of HAuCl4 (0.1 M) was added to 2 mL of the supernatant and incubated overnight on a shaker incubator at 200 rpm at 37 °C. In order to chemically synthesize the gold nanoparticles (Turkevich method [21–25]), 1.5 mL trisodium citrate was immediately added to 15 mL of the boiling chloroauric acid solution. After the appearance of the red color, a solution containing gold nanoparticles was placed at room temperature to cool (Figure 9).

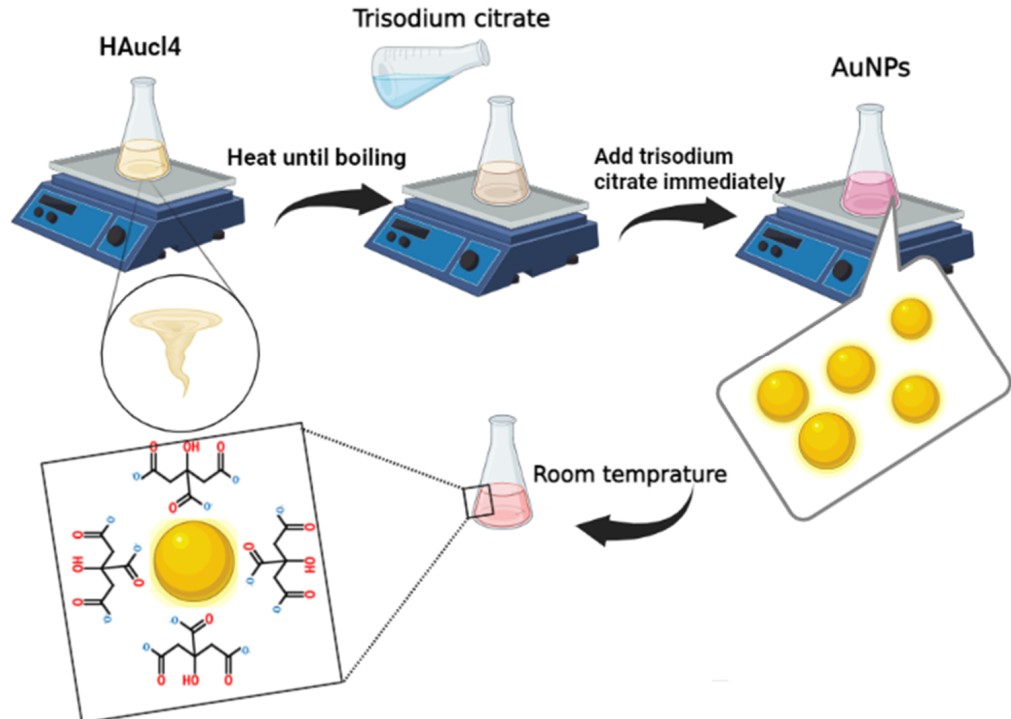

**Figure 9.** Schematic of gold nanoparticle chemical synthesis.

### 4.2. Characterization of Gold Nanoparticles

The characterization of gold nanoparticles was performed by X-Ray diffraction (XRD) (StoE, Hilpertstrasse, Darmstadt, Germany), atomic force microscopy (AFM)(VEECO instrument, Plainview, NY, USA) and field-emission scanning electron microscopy (FESEM). The morphology and size of nanoparticles were examined by the FESEM method (TESCAN, Brno, Czech Republic). XRD was conducted to analyze the crystalline structure of nanoparticles at an angle of ($2\theta$) = 35 to 80° with Cu Ka radiation at A = 1.5406 Å.

### 4.3. Antibody Synthesis

The antibodies were synthesized according to the method described by Yarian et al. [26].

### 4.4. Binding of Gold Nanoparticles to Antibodies

In this experiment, 1 mL of PBS (1X) was added to 25 mL of the gold nanoparticle solution; then, 500 μL of the antibody (scFv) solution (1 μg/mL) was added dropwise to the nanoparticle solution and placed at 4 °C for 48 h. Unbound antibodies were then removed by centrifugation (12,000 rpm, 15 min, three times), and the pellet was dissolved in 2 mL of PBS and stored at 4 °C until use.

### 4.5. Agarose Gel

The antibody-bonded solution of gold nanoparticles was centrifuged at 12,000 rpm for 15 min. Then, the pellet was loaded on 1% agarose gel to assess the binding of gold nanoparticles to the antibody.

### 4.6. UV-Vis Spectroscopy

After 24 h of adding the antibodies to the gold nanoparticles and removing the un-bound antibodies, the conjugate solution of gold nanoparticles and antibodies dissolved in PBS (1X) buffer was analyzed by UV-Vis spectroscopy (Lambda 25, Perkin Elmer, MA, USA).

### 5. Conclusions

The results showed that chemically synthesized gold nanoparticles are able to bind to the scFv antibody due to the surface charge of citrate existing on the surface of these nanoparticles. Interestingly, this experiment was carried out using the recombinant scFv antibodies that do not have a complete antibody structure. Thus, further investigations are necessary to analyze whether this method can be applied to structurally complete antibodies and other types of linkers. The biological pathway of nanoparticle synthesis is not understood completely. It is hoped that with future developments and accurate identification of the capping and stabilizing agents on the surface of biosynthesized nanoparticles, these nanoparticles can be effectively used in functionalization processes.

**Author Contributions:** M.R., M.B., G.E. and O.A. collaborated in the practical part, and also, M.R., M.B., G.E., O.A. and F.Y. participated in the final article. All authors have read and approved the final version of the manuscript.

**Funding:** The project was funded by [the Shahid Beheshti University of Medical Sciences] grant number [26266] Code of Ethics [IR. SBMU.RETECH.REC.1400.080].

**Data Availability Statement:** Data is contained within the article.

**Acknowledgments:** The authors would like to thank the Shahid Beheshti University, Shahid Beheshti University of Medical Sciences and the Cellular and Molecular Biology Research Center of Shahid Beheshti University of Medical Sciences for supporting this research project.

**Conflicts of Interest:** The authors declare no conflict of interest.

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
