# Peer review of "Enzymatic Formation of Recombinant Antibody-Conjugated Gold Nanoparticles in the Presence of Citrate Groups and Bacteria"

_catalysts, doi:10.3390/catal12091048_

Round 1
Reviewer 1 Report
The research work entitles "Enzymatic formation of recombinant antibody-conjugated gold nanoparticles in the presence of bacteria" was submitted by Maryam Rad et al. Overall the manuscript is good. There are few important corrections to be carried out for improving the quality of manuscript.
1. all the scientific name of microbes should be italics.
2. Escherichia. Coli should be corrected as Escherichia coli.
3. Include the size and shape of the nanoparticles in abstract.
4. Expand N.B.
5. Provide collection details of Escherichia. coli TOP-10.
Author Response
Thanks for the valuable and constructive comments of the reviewers about the work. We tried to revise the manuscript based on the comments of the reviewer as follows:
- all the scientific name of microbes should be italics.
Response: It has been done and marked by red color in the text.
- Coli should be corrected as Escherichia coli.
Response: It is completely right. It has been done and marked by red color in the text.
- Include the size and shape of the nanoparticles in abstract.
Response: Our result showed that gold nanoparticles had spherical morphology and their average size was ~45 nm. This has been mentioned in the revised version, as requested by the reviewer.
- Expand N.B.
Response: Thanks. It is defined as follows: nutrient broth (N.B.)
- Provide collection details of Escherichia coli TOP-10
Response: The Catalog number C404010 has been mentioned in the revised version, based on the comment of the reviewer.
Reviewer 2 Report
Some minor modifications are given for catalysts-1923247 in the attached file; overall, paper is good and merits publication in the journal.

Author Response
Thanks for the valuable and constructive comments of the reviewers about the work. We tried to revise the manuscript based on the comments of the reviewer as follows:
- Title may be improved.
Response: Thanks for the right comment. The tile has been improved to “Enzymatic formation of recombinant antibody-conjugated gold nanoparticles in the presence of citrate groups and bacteria”, as requested by the reviewer.
- Line 95 change adsorption to absorption
Response: It has been done and marked by red color in the text.
- Neither Figure 4 nor its explanation is appealing.
Response: Figure 4 has been replaced with another one having better quality. It explanation has been improved, as mentioned below:
Figure 4. Illustrated immigration pattern of Au NPs in various conditions: 1) the chemical synthesized Au NPs, 2) the scFv-conjugated Au NPs prepared by the chemical method, 3) the biological synthesized Au NPs, and 4) the scFv-conjugated Au NPs prepared by the biological method. For the chemical method, the immigration pattern of Au NPs has changed, while in biological method the patterns are the same before and after functionalization.
- in Figure 5, absorbance shift moved > 550 nm, indicate the difference between peaks of chemically synthesized and biologically synthesized ones.
Response: Gold nanoparticles have an absorption peak in the range of 500 to 600 nm. Since the synthesis method was different (different solutions in the reaction medium), the absorption peak pattern is slightly different, but it is still in the range of 500 to 600 nm. In this picture, the aim is to compare the absorption peak pattern of nanoparticles before binding to the antibody and after binding to it (for example, both naked AuNPs and conjugated nanoparticles should be in a chemical method, not one of them in chemical and another in biological). Therefore, comparing the absorption peak changes in two different synthesis methods is not the ambition of this research. Nevertheless, regarding the difference in the stabilizing agent of chemical and biological synthesis nanoparticles, the difference in absorption peaks is also justified.
- In Conclusions section, just results for the chemically synthesized AuNPs has been concluded.
Response: The biological pathway of nanoparticle synthesis is not understood completely. It is hoped that with future developments and accurate identification of the capping and stabilizing agents on the surface of biosynthesized nanoparticles, these nanoparticles can be effectively used in functionalization processes. Now, this statement has been added to the conclusion section, as commented by the reviewer.